# Distinct Chemokine Receptor Expression Profiles in De Novo DLBCL, Transformed Follicular Lymphoma, Richter’s Trans-Formed DLBCL and Germinal Center B-Cells

**DOI:** 10.3390/ijms23147874

**Published:** 2022-07-17

**Authors:** Barbara Uhl, Katharina T. Prochazka, Katrin Pansy, Kerstin Wenzl, Johanna Strobl, Claudia Baumgartner, Marta M. Szmyra, James E. Waha, Axel Wolf, Peter V. Tomazic, Elisabeth Steinbauer, Maria Steinwender, Sabine Friedl, Marc Weniger, Ralf Küppers, Martin Pichler, Hildegard T. Greinix, Georg Stary, Alan G. Ramsay, Benedetta Apollonio, Julia Feichtinger, Christine Beham-Schmid, Peter Neumeister, Alexander J. Deutsch

**Affiliations:** 1Division of Hematology, Department of Internal Medicine, Medical University of Graz, 8036 Graz, Austria; barbara.uhl@medunigraz.at (B.U.); katharina.prochazka@uniklinikum.kages.at (K.T.P.); katrin.pansy@medunigraz.at (K.P.); wenzl.kerstin@majo.edu (K.W.); marta.szmyra@medunigraz.at (M.M.S.); hildegard.greinix@medunigraz.at (H.T.G.); 2Division of Hematology, Mayo Clinic, Rochester, MN 55902, USA; 3Department of Dermatology, Medical University of Vienna, 1090 Vienna, Austria; johanna.strobl@meduniwien.ac.at (J.S.); georg.stary@meduniwien.ac.at (G.S.); 4Division of Cell Biology, Histology and Embryology, Gottfried Schatz Research Center, Medical University of Graz, 8036 Graz, Austria; claudia.baumgartner@medunigraz.at (C.B.); julia.feichtinger@medunigraz.at (J.F.); 5General, Visceral and Transplant Surgery, Medical University of Graz, 8036 Graz, Austria; jameselvis.waha@klinikum.kages.at; 6Department of Otorhinolaryngology, Head and Neck Surgery, Medical University of Graz, 8036 Graz, Austria; a.wolf@medunigraz.at (A.W.); peter.tomazic@medunigraz.at (P.V.T.); 7Institute of Pathology, Medical University of Graz, 8036 Graz, Austria; elisabeth.steinbauer@uniklinikum.kages.at (E.S.); maria.steinwender@uniklinikum.kages.at (M.S.); sabine.friedl@uniklinikum.kages.at (S.F.); christine.beham@medunigraz.at (C.B.-S.); 8Institute of Cell Biology (Cancer Research), University of Duisburg-Essen, 45122 Essen, Germany; marc.weniger@uk-essen.de (M.W.); ralf.kueppers@uk-essen.de (R.K.); 9German Cancer Consortium (DKTK), 69120 Heidelberg, Germany; 10Division of Oncology, Department of Internal Medicine, Medical University of Graz, 8036 Graz, Austria; martin.pichler@medunigraz.at; 11Ludwig Boltzmann Institute for Rare and Undiagnosed Diseases, 1090 Vienna, Austria; 12CeMM Research Center for Molecular Medicine of the Austrian Academy of Sciences, 1090 Vienna, Austria; 13Faculty of Life Sciences & Medicine, School of Cancer & Pharmaceutical Sciences, King’s College London, London WC2R 2LS, UK; alan.ramsay@kcl.ac.uk (A.G.R.); benedetta.apollonio@kcl.ac.uk (B.A.)

**Keywords:** diffuse large B-cell lymphoma, Richter syndrome, transformed follicular lymphoma, chemokine receptors, CCR7, CCL19, CCL21

## Abstract

Chemokine receptors and their ligands have been identified as playing an important role in the development of diffuse large B-cell lymphoma (DLBCL), follicular lymphoma, and Richter syndrome (RS). Our aim was to investigate the different expression profiles in de novo DLBCL, transformed follicular lymphoma (tFL), and RS. Here, we profiled the mRNA expression levels of 18 chemokine receptors (*CCR1*–*CCR9*, *CXCR1*–*CXCR7*, *CX3CR1* and *XCR1*) using RQ-PCR, as well as immunohistochemistry of seven chemokine receptors (CCR1, CCR4–CCR8 and CXCR2) in RS, de novo DLBCL, and tFL biopsy-derived tissues. Tonsil-derived germinal center B-cells (GC-B) served as non-neoplastic controls. The chemokine receptor expression profiles of de novo DLBCL and tFL substantially differed from those of GC-B, with at least 5-fold higher expression of 15 out of the 18 investigated chemokine receptors (*CCR1*–*CCR9*, *CXCR1*, *CXCR2*, *CXCR6*, *CXCR7*, *CX3CR1* and *XCR1*) in these lymphoma subtypes. Interestingly, the de novo DLBCL and tFL exhibited at least 22-fold higher expression of *CCR1*, *CCR5*, *CCR8*, and *CXCR6* compared with RS, whereas no significant difference in chemokine receptor expression profile was detected when comparing de novo DLBCL with tFL. Furthermore, in de novo DLBCL and tFLs, a high expression of CCR7 was associated with a poor overall survival in our study cohort, as well as in an independent patient cohort. Our data indicate that the chemokine receptor expression profile of RS differs substantially from that of de novo DLBCL and tFL. Thus, these multiple dysregulated chemokine receptors could represent novel clinical markers as diagnostic and prognostic tools. Moreover, this study highlights the relevance of chemokine signaling crosstalk in the tumor microenvironment of aggressive lymphomas.

## 1. Introduction

Diffuse large B-cell lymphoma (DLBCL) is the most common lymphoma subtype, accounting for 30–40% of all lymphomas in adults with a 5-year survival rate of around 50% [1]. Although contemporary chemo-immunotherapy can cure about half of these patients, there is an urgent medical need to improve actual therapy regimens [2]. Based on gene expression profiling, DLBCL can be divided into two different subtypes: (i) germinal center B-cell-like (GCB-DLBCL); and (ii) activated B-cell-like (ABC-DLBCL), or, in the case of the usage of an immunohistochemical algorithm, non-germinal center B-cell-like (NGCB-DLBCL) [3]. DLBCL arises de novo or by the transformation of indolent lymphomas, such as follicular lymphomas (FL), and chronic lymphocytic leukemia (CLL), so-called Richter syndrome (RS). About 2–3% of the patients with FL and 0.5–1% of the patients with CLL or small lymphocytic leukemia transform per year [4,5]. Transformed FL (tFL) and RS-derived DLBCL are characterized by a more aggressive clinical course compared with the de novo DLBCL [6,7]. Especially in RS, only a limited number of patients achieve long-term survival and, therefore, have an extremely poor outcome [6].

Chemokine receptors bind chemokines that are also known as pro-inflammatory and chemotactic cytokines. The chemokines, as well as the chemokine receptors, play a key role in diverse biological functions, e.g., organogenesis, hematopoiesis, and inflammatory processes [8,9,10]. CCR6, CCR7, CXCR3, CXCR4, and CXCR5 are crucial for B-cell development, and, therefore, they are referred as B-cell homeostatic chemokine receptors [11,12,13]. The activation-induced chemokine receptors, which are expressed in response to the inflammatory cytokines, exert an essential role in inflammation and are responsible for the migration of the immune cells toward a chemokine gradient, produced by the inflamed cells [8]. Furthermore, it has been demonstrated that the chemokine receptors are involved in the development, dissemination, and progression of B-cell lymphoma [14,15,16,17,18]. The development of the chemokine antagonists, and their use in clinical trials for a variety of diseases, show the potential of the chemokine receptors for molecular-targeted therapy [19].

The present study investigated the expression patterns of 18 chemokine receptors of de novo DLBCL, tFL, RS, and germinal center B-cells (GC-B). Here, we show that 15 of the 18 investigated chemokine receptors were differentially expressed in GCB-DLBCL, NGCB-DLBCL, and tFL, and eight chemokine receptors in the RS subgroup. Interestingly, *CCR1*, *CCR5*, *CCR8*, *CXCR6*, and *CX3CR1* were more highly expressed in de novo DLBCL and tFL compared with RS, whereas no differences were observed when comparing GCB-DLBCL, NGCB-DLBCL, and tFL to each other. Interestingly, a high expression of CCR7 was associated with poor overall survival.

## 2. Results

### 2.1. Substantial Differences in the CCR and CXCR Expression Patterns in DLBCL, tFL, RS and GC-B

Since the knowledge of the chemokine receptor expression profiles in de novo and transformed DLBCL is limited, we examined the mRNA expression levels of 18 well-characterized chemokine receptors in primary lymphoma tissue samples, consisting of GCB-DLBCL (*n* = 8) and NGCB-DLBCL (*n* = 18), one unclassified de novo DLBCL, tFL (*n* = 16), and RS (*n* = 14), as well as GC-B (*n* = 4) serving as the non-neoplastic controls.

The activation-induced chemokine receptors [8], *CCR2*, *CCR3*, and *CCR8*, showed a significantly higher expression in GCB-DLBCL, NGCB-DLBCL, tFL, and RS compared with GC-B. Remarkably, these three CCR receptors, as well as *CCR1*, *CCR4*, and *CCR5*, three other activation-induced chemokine receptors [8], showed at least a 29-fold higher expression in GCB-DLBCL, NGCB-DLBCL, and tFL compared with GC-B (Figure 1, *p* < 0.033). In RS, we detected at least a 23-fold higher expression of *CCR2*, *CCR3*, and *CCR8* compared with GC-B (Figure 1, *p* < 0.012). *CCR9*, an additional member of the activation-induced chemokine receptor family [8], showed a higher expression (at least 21-fold) in NGCB-DLBCL and tFL compared with controls (Figure 1, *p* < 0.014). Regarding the two members of the B-cell homeostatic chemokine receptors [11,12,13], *CCR6* and *CCR7*, we observed a 7-fold higher *CCR6* expression in tFL (Figure 1, *p* = 0.038), and at least a 5-fold higher *CCR7* expression in GCB- and NGCB-DLBCL, tFL, and RS compared with controls (Figure 1, *p* < 0.028).

In the group of CXCR, CX3CR1, and XCR1 chemokine receptors, five activation-induced chemokine receptors [8], *CXCR1*, *CXCR2*, *CXCR6*, *CX3CR1*, and *XCR1* were expressed at least 27-fold more highly in GCB-DLBCL, NGCB-DLBCL, and tFL, compared with GC-B (Figure 2, *p* < 0.043). In RS, we detected an 18-fold higher expression of *CXCR2*, *CX3CR1*, and *XCR1* compared with GC-B (Figure 2, *p* < 0.048). Interestingly, we did not observe any substantial differences between GC-B and the lymphoma entities for the B-cell homeostatic chemokine receptors, *CXCR3*, *CXCR4*, and *CXCR5* [11,12,13] (Appendix A, Appendix A).

Comparing the chemokine receptor expression profiles between the different lymphoma subtypes, we observed a remarkably higher expression of *CCR1*, *CCR5*, *CCR8*, *CXCR6*, *CXCR7*, and *CX3CR1* (42-fold, 123-fold, 51-fold, 20-fold, 21-fold, and 4-fold higher expression) in both DLBCL-subtypes and tFL compared with RS (Figure 1 and Figure 2, *p* < 0.045). Interestingly, no difference in the chemokine receptor expression patterns was detected comparing between GCB-DLBCL, NGCB-DLBCL, and tFL.

To further explore the chemokine receptor expression patterns of the lymphoma subtypes, we performed hierarchical clustering based on the mRNA expression levels using ΔCT-values. As shown in Figure 3, the samples can be delineated as two discrete clusters, based on their expression profiles. The first cluster contained all of the GC-B (4/4) and most of the RS (11/14) samples, whereas the second cluster comprised all of the DLBCL (8/8 GCB-DLBCLs, 18/18 NGCB-DLBCLs, and 1/1 unclassified DLBCL) and tFL (16/16) samples, as well as a few RS (3/14) samples. In concordance with the differential expression analysis results, these data suggest that the vast majority of the investigated specimens with RS differ in their chemokine receptor expression profiles from the de novo DLBCL and tFL and have an expression profile more similar to GC-B than to other lymphoma subtypes.

### 2.2. The Chemokine Receptor Expression Pattern Is not Influenced by Intratumoral Inflammatory Cells

To decipher the influence of the lymphoma-infiltrating immune cells on the chemokine receptor expression profiles, we analyzed the intratumoral T-cells (CD3^+^), T helper (CD3^+^CD4^+^), and cytotoxic T (CD3^+^CD8^+^) subpopulations determined by multicolor-immunofluorescence staining (*n* = 20), as well as the macrophages (CD68^+^) by immunohistochemistry (IHC) (*n* = 31) on selected cases, which were used for the mRNA expression profiling, as shown in Table 1 and Appendix A (Appendix A). No correlation between the number of intratumoral T-cells or their subpopulations, macrophages, and the chemokine receptor profile was observed, suggesting that the chemokine receptor profile might be determined by the presence of lymphoma B-cells and not by the infiltrating non-malignant immune cells. Interestingly, the CD68 content was higher in NGCB-DLBCL compared with tFL (*p* < 0.039).

### 2.3. Reduced CCR5, CCR6, and CCR8 Protein Content in RS Compared with De Novo DLBCL and tFL

Owing to the highest mRNA expression in the de novo DLBCL and tFL lymphoma subtypes compared with GC-B and RS, immunohistochemical analyses of the five chemokine receptors (CCR1, CCR4, CCR5, CCR6, and CCR8) were performed on selected cases (*n* = 36 in total: 16 de novo DLBCL, 12 tFLs, and 8 RS, identical to the ones of the mRNA expression analyses). As the controls, reactive tonsils (*n* = 4) from non-neoplastic donors were included to evaluate the reactive GC-B. When comparing the protein data with mRNA expression, a significant positive correlation of CCR5 (Spearman-rho 0.519, *p* = 0.027); CCR6 (Spearman-rho 0.556, *p* = 0.031), and CCR8 (Spearman-rho 0.671, *p* = 0.005) was observed (Figure 4), thereby indicating that the deregulated mRNA expression translates into abnormal protein content. For the CCR1 and CCR4, no correlation was found (Appendix A, Appendix A), which suggests that the mRNA expression does not translate into protein content in the investigated lymphoma subgroups, and that these two receptors are regulated at post-translational levels.

In the reactive tonsils, all of the GC-Bs strongly expressed CCR5 on their surface. A total of 80% of the centrocytes and 80% or 50% of the centroblasts strongly expressed CCR6 and CCR8. In contrast, neither of the two cell types expressed CCR1 or CCR4 (Appendix A, Appendix A).

In the de novo DLBCL specimens, the majority of the GCB- and NGCB-DLBCL cells strongly expressed CCR5 and CCR8 (Table 2 and Figure 4). Additionally, a moderate expression of CCR6 was found on the majority of lymphoma cell surfaces in both of the molecular subtypes (Table 2 and Figure 4). In contrast, missing or weak CCR1 and CCR4 expression was detected in a small fraction of the malignant cells (Appendix A, Appendix A).

The tFL specimens possess a similar protein pattern to the de novo DLBCL: CCR5 and CCR8 were strongly and CCR6 was moderately expressed, whereas CCR1 and CCR4 expression was rarely or weakly detectable on the malignant cells (Table 2 and Figure 4; Appendix A, Appendix A).

In the group of RS, CCR5 and CCR8 were strongly and CCR6 was moderately expressed on the majority of the malignant cells. Interestingly, the percentage of the lymphoma cells stained positively for CCR5 and CCR8 was lower in the RS samples compared with GCB-, NGCB-DLBCL, and tFL (90% CCR5^+^ cells in GCB- and NGCB-DLBCL and 83.3% for tFL vs. 70% for RS, *p* < 0.1) confirming the RNA data. As in the other lymphoma entities, the CCR1 and CCR4 expression was rarely or weakly detectable (Table 2 and Figure 4; Appendix A, Appendix A).

### 2.4. High CCR7 Expression Is Associated with Poor Overall Survival and High CXCR2 Content in De Novo DLBCL and tFL

To investigate the clinical relevance of the generated expression data, we exploratively set the expression of the 18 chemokine receptors in relation to overall survival. For this purpose, our de novo DLCBL and tFL samples (*n* = 43 in total) were selected and analyzed as one group, because of their similar chemokine receptor expression profiles and the fact that all of the patients received R-CHOP-like therapy [20]. Because of our small sample size, we used the microarray data published by Lenz et al. as an additional independent cohort (*n* = 200 of patients treated with R-CHOP and assigned to the subtype ABC-DLBCL or GCB-DLBCL [21]). By using the median of the mRNA expression, the patients were divided into two groups, and we observed that a high expression of *CCR7* was associated with poor overall survival in our cohort (*p* = 0.013, Figure 5a) as well as in the Lenz et al. dataset (*p* = 0.0016, Figure 5b). None of the other chemokine receptors had a significant association with survival in both of the cohorts.

To determine whether a high expression of *CCR7* translated into high or low protein levels, an immunohistochemical analysis for this chemokine receptor was performed. A positive correlation was observed for CCR7 (Spearman-rho 0.698, *p* = 0.005, Table 3 and Figure 6), validating the mRNA data. Interestingly, CCR7 was moderately expressed on the majority of the lymphoma cells of GCB- and NGCB-DLBCL and tFL (Table 3 and Figure 6). In the reactive tonsils, all of the GC-Bs exhibited strong CCR7 protein expression (Figure 6).

To further comprehensively study CCR7 in de novo DLBCL and tFL, we immunohistochemically analyzed CCL19 and CCL21, the ligands of CCR7 [22,23,24,25,26,27], in our de novo DLBCL and tFL cohorts (*n* = 43). Moderate CCL19 expression was detectable on the vast majority of the lymphoma cells in both of the DLBCL subtypes (Table 3 and Figure 6), whereas CCL21 protein expression was found in less than half of the analyzed samples (18 of 43). In the reactive tonsils, all of the centrocytes and centroblasts moderately or strongly expressed the CCL19 and CCL21 proteins (Figure 6). Combining the CCR7 expression with the CCL19 and CCL21 expression pattern, it seems that CCR7 signaling is altered in the lymphoma setting compared to the non-neoplastic condition. This potentially results in the association of high CCR7 with a poor outcome.

Owing to the highest *CXCR2* mRNA expression in the de novo DLBCL and tFL lymphoma subtypes compared with GC-B, an immunohistochemical analysis of this chemokine receptor was performed. A positive correlation between the mRNA and protein levels was observed for CXCR2 (*n* = 28, Spearman-rho 0.696, *p* = 0.002, Table 3 and Figure 6), validating the mRNA data. In the reactive tonsils (*n* = 4), all of the GC-Bs possessed a weaker CXCR2 protein expression compared with the lymphoma cells (Figure 6).

## 3. Discussion

This study was designed to exploratively investigate the mRNA expression pattern of the chemokine receptors in de novo DLBCL, tFL, and RS, since it was demonstrated that chemokine receptors play an important role in lymphomagenesis and most of them were not so far intensively investigated in GCB-DLBCL, NGCB-DLBCL, tFL, and RS.

Although we used a small number of the lymphoma samples of the different subgroups, we reported that all of the investigated activation-induced chemokine receptors –*CCR1–CCR5*, *CCR8*, *CCR9*, *CXCR1*, *CXCR2*, *CXCR6*, *CXCR7*, *CX3CR1*, and *XCR1*—as well as two of the five investigated B-cell homeostatic chemokine receptors –*CCR6* and *CCR7*—were more highly expressed in at least NGCB-DLBCL, GCB-DLBCL, and tFL compared with GC-B. In RS, we detected a higher expression of *CCR2*, *CCR3*, *CXCR1*, *CXCR2*, *CX3CR1*, and *XCR1*, which were also more highly expressed in the three other DLBCL subgroups. The expression of all of these chemokine receptors has already been described in DLBCL but rarely in RS [9,12,15,17,28,29,30,31,32,33,34,35,36,37]. Our analysis and published data indicate that the chemokine receptors might play a key role in the pathogenesis of de novo and transformed DLBCL.

In our comprehensive chemokine receptor expression analysis, we also found that the expression profile of RS substantially differs from the expression profile of GCB-DLBCL, NGCB-DLBCL, and tFL, with a lower expression of *CCR1*, *CCR5*, *CCR8*, *CXCR6*, *CXCR7*, and *CXCR3* in RS. This is potentially caused by the fact that the investigated DLBCL subgroups originate at different stages of the B-cell differentiation. RS develops in the context of a CLL, which is derived either from mature CD5^+^ B-cells or memory B-cells, whereas GCB-DLBCL and tFL arise from GC-B and NGCB-DLBCL from post-GC-B [38,39,40]. Similarly, our cluster analysis suggests differences in the chemokine receptor expression profile of RS compared with the different lymphoma groups. Interestingly, the majority of the RS group clustered together with the GC-B cells. This might be caused by the acquired genomic alteration of the RS samples [5,41,42,43], which results in autonomous lymphoma cell growth.

Our analysis of the content of the lymphoma infiltration immune cells (CD3^+^CD4^+^-, CD3^+^CD8^+^-, and CD68^+^-cells), is similar to the study of Autio et al. [44]. Furthermore, we observed a low percentage of the CD68^+^ cells in tFL compared with NGCB-DLBCL, indicating a different function of the lymphoma-infiltrating immune cells in the different subgroups. Furthermore, the fact that we did not observe any association between the immune cell content and chemokine receptor expression profile, indicates that the expression pattern was determined by the lymphoma cells and not by the immune cells.

Our immunohistochemical analysis demonstrated moderate to strong expression of CCR5, CCR6, and CCR8 on the surface of the lymphoma cells in all of the investigated subgroups. It was demonstrated that the interaction of these three receptors plays a key function in the proliferation of malignant cells [45]. Thus, we hypothesize that CCR5, CCR6, and CCR8 might play a major role in the pathogenesis of GCB-DLBCL, NGCB-DLBCL, tFL, and RS.

We also detected high CCR7 expression on the lymphoma cells of the GCB-DLBCL, NGCB-DLBCL, and tFL subgroups. Our survival analysis revealed that high CCR7 expression is associated with a poor prognosis, which we also confirmed in the dataset of Lenz et al. [21], serving as an additional independent cohort. This observation is in accordance with the study of Du et al. [35], who showed a similar association in primary nodal DLBCL. Additionally, it was shown that CCR7 plays a key role in the homing of the tumor cells into the lymphoma-supporting niches in aggressive murine B-cell lymphomas [46]. Our immunohistochemical analysis of the CCL19 and CCL21-ligands of CCR7 [22,23,24,25,26,27] demonstrated that the lymphoma cells expressed CCL19, but that CCL21 expression was low in the malignant cells. It was shown that both of the ligands can activate CCR7 signaling to deliver survival and proliferation signals, as well as to cause chemotaxis, but CCL19 has been shown to be more potent in this regard than CCL21 [47]. Thus, taking our current and published data together, we hypothesize that autocrine CCR7-CCL19 signaling may significantly contribute to lymphomagenesis under malignant conditions by a stronger activation of the survival pathways.

Furthermore, in our cohort, we did not observe any association between the *CXCR4* expression level and survival, which is contradictory to the studies reported by us [18], and by other research groups [48,49]. This might be caused by the small sample size of our cohort and differences in the included patients.

Our immunohistochemical analysis of CXCR2 revealed a strong expression on the de novo DLBCL and tFL cells. Expression of this chemokine receptor has been reported on low grade FL, small lymphocytic lymphoma, marginal zone lymphoma, and hairy cell leukemia [15], but has never been investigated in DLBCL so far. Since it has been reported that the DLBCL cells produce CXCL8 [50], the ligand of CXCR2 [51], we speculate that the CXCL8–CXCR2-axis might play a key role in lymphomagenesis.

## 4. Materials and Methods

### 4.1. Patients

To comprehensively study the expression of the chemokine receptors in the development of aggressive lymphomas, we measured the mRNA expression of 18 well-characterized chemokine receptors in our cryopreserved aggressive lymphoma tissue samples (*n* = 57). The clinicopathologic parameters of the lymphoma cohort are shown in Table 4; the patients were treated at the Division of Hematology, Medical University of Graz between 2000 and 2010 (with the last follow-up until 2020). Following the World Health Organization classification [52], 27 of the samples were classified as de novo DLBCL and 30 secondary DLBCL originating from tFL (*n* = 16) and RS (*n* = 14). By using the Hans algorithm [3], the 27 de novo DLBCL samples were subdivided into 8 GCB-DLBCL subtypes and 18 NGCB-DLBCL subtypes, respectively. Due to lacking tumor material, the classification of one lymphoma sample was not possible.

For this retrospective study, we used patient specimens obtained during routine diagnostic procedures. Hence, no written informed consent from patients was obtained. The study was conducted in accordance with the Declaration of Helsinki, and the protocol was approved by the Ethics Committee of the Medical University of Graz (ethical application 28-516 ex 15/16).

Non-neoplastic GC-Bs (CD20^high^CD38^+^) serving as control were isolated using a FACS-ARIA3 cell sorter from four human tonsils collected from children and adolescents undergoing routine tonsillectomy with the donor’s informed consent, as approved by the Ethics Committee of the Medical School of the University Duisburg-Essen (number of ethical application: 11-4799-30).

### 4.2. RNA Isolation, cDNA Synthesis and Real-Time PCR

The RNA isolation was performed using the miRNeasy Mini Kit (Qiagen, Hilden, Germany), according to the manufacturer’s protocol. The isolated RNA was then transcribed into cDNA using the RevertAid H Minus First Strand cDNA Synthesis Kit (ThermoFisherScientific, Waltham, MA, USA), according to the manufacturer’s protocol.

Semi-quantitative real-time PCR (RQ-PCR) was performed in triplicates using the Bio-Rad CFX 384 detection system (Bio-Rad, Hercules, CA, USA). The nucleotide acid sequences for primers and probes used for RQ-PCR analysis of 18 chemokine receptors (*CCR1*-*CCR9*, *CXCR1*-*CXCR7*, *CX3CR1*, and *XCR1*) are shown in Appendix A (Appendix A). GAPDH and PPIA, possessing a high correlation coefficient (Spearman rho > 0.85 and *p*  <  0.05), served as the housekeeping genes. All of the PCR assays were already tested, validated, and published in previous studies [17,53]. The results are expressed as relative units based on calculation 2^−ΔΔCT^, giving the relative amount of the target gene normalized to the endogenous control (geometric mean of the GAPDH and PPIA) and relative to peripheral mononucleated cells [54,55]. The undetermined values were set to a maximum of CT 45, in accordance with that of Goni et al. [55]

### 4.3. Immunohistochemical Analyses for CCR1, CCR4, CCR5, CCR6, CCR7, CCR8, CXCR2, CCL19, and CCL21

Formalin-fixed, paraffin-embedded tissue was pretreated in a water bath with Target Retrieval Solution (1:10, Dako, Glostrup, Denmark) for 40 min. The primary antibodies and dilutions for all of the investigated proteins are listed in Appendix A (Appendix A). For staining, kit K5001 (Dako, Glostrup, Denmark) and the automated stainer intelliPATH FLX^®^ (Biocare Medical, Pacheco, CA, USA) were used, according to the manufacturer’s instructions. We included tissues known to contain the respective antigens—reactive tonsils—as controls (positive controls). Replacing the primary antibody with normal serum always produced negative results (negative controls).

For scoring, the percentage of positive cells and the intensity of staining were graded from 0 to 2+: 0, no staining; 1+, weak positive staining; 2+, moderate to strong positive staining. The determination of the percentage of positively stained cells was performed by calculating the average percentage of positive cells in at least ten high-power-fields (HPFs) (0.242 mm^2^ each, field diameter: 555.1 µm). The percentages were rounded to 10%. An immunoreactive score (IRS) was obtained by multiplying the percentage of positive cells by staining intensity divided by 10, according to Zhuang et al. [56].

### 4.4. Multicolor-Immunofluorescence Staining of T-Cells and Subpopulations

Quadruple immunofluorescence staining of the acetone-fixed cryosections was performed using directly labeled monoclonal and secondary antibodies for increased signal strength (Appendix A, Appendix A). Briefly, after rehydration and blocking, the slides were incubated in several steps with primary antibodies overnight at 4 °C and appropriate secondary antibodies for 30 min at room temperature, followed by counterstaining with DAPI. Appropriate isotype controls were stained at the same time. For evaluating the immunofluorescence results, the images were acquired at room temperature using a Z1 Axio Observer microscope equipped with an LD Plan-Neofluar ×20/0.4 objective (Zeiss, Oberkochen, Germany) using TissueFAXS imaging software and quantified with TissueQUEST image analysis software (TissueGnostics, Vienna, Austria).

### 4.5. Statistics

Statistical analysis was performed using SPSS 25.0 (SPSS Inc, Chicago, IL, USA). The chemokine receptor expression levels with significant differences in their expression were analyzed using the Mann–Whitney U-test; all of the significant associations were further corrected for multiple testing by applying a Bonferroni correction, dividing the significance level by the number of variables tested.

### 4.6. Survival Analysis and Heatmap

ΔCT-values were used to generate the heatmap, similarly to previously described [57,58,59]. The heatmap and hierarchical clustering with Euclidean distance and Ward linkage were created using the heatmap.2 function of the R package ‘gplots’ [60] (R version 3.6.3. [61]), applied to scaled data.

For the survival analysis, we analyzed the mRNA expression values of 18 well-characterized chemokine receptors and the clinical data of 43 lymphoma patients, suffering from de novo DLBCL and tFL (summarized in Table 4). The patients were split into two groups using the median of mRNA expression values for each chemokine receptor. Survival analysis was carried out using the R package ‘survival’ [62] (R version 3.6.3. [61]. Overall survival was evaluated using the Kaplan–Meier method and compared by the log-rank test. The resulting survival curves were plotted using the R package ‘survminer’ [63]. *p*-values ≤0.05 were considered statistically significant. Only the subjects with complete survival data available were used in the analysis. To reproduce our findings in a public dataset, we re-analyzed the microarray data published by Lenz et al. (E-GEOD-10846 dataset available from ArrayExpress) [21]. The raw data were processed and normalized, using the ‘oligo’ R package [64] (R version 3.5.1 [61]). We incorporated the patient samples treated with R-CHOP and classified as GCB-DLBCL or ABC-DLBCL in the analysis. Survival analysis was performed as described above. Due to the low sample size of our cohort, only the results for chemokine receptors that were significant in both of the cohorts were presented here.

## 5. Conclusions

In conclusion, we demonstrated that only two of the B-cell homeostatic chemokine receptors and the majority of the activation-induced chemokine receptors showed a higher expression in GCB-DLBCL, NGCB-DLBCL, tFL, and RS lymphoma samples compared with non-malignant GC-B. We showed substantial differences in the chemokine receptor profile of RS compared to that of GCB-DLBCL, NGCB-DLBCL, and tFL. Additionally, we demonstrated that the expression levels of CCR7 could be of prognostic relevance and, finally, that de novo DLBCL and tFL might exhibit autocrine CCR7-CCL19 signaling. Therefore, (a combination of) chemokine receptors might serve as useful predictive or diagnostic tools and might represent targets for novel therapeutic approaches of de novo DLBCL, tFL, and RS.

## Figures and Tables

**Figure 1 ijms-23-07874-f001:**
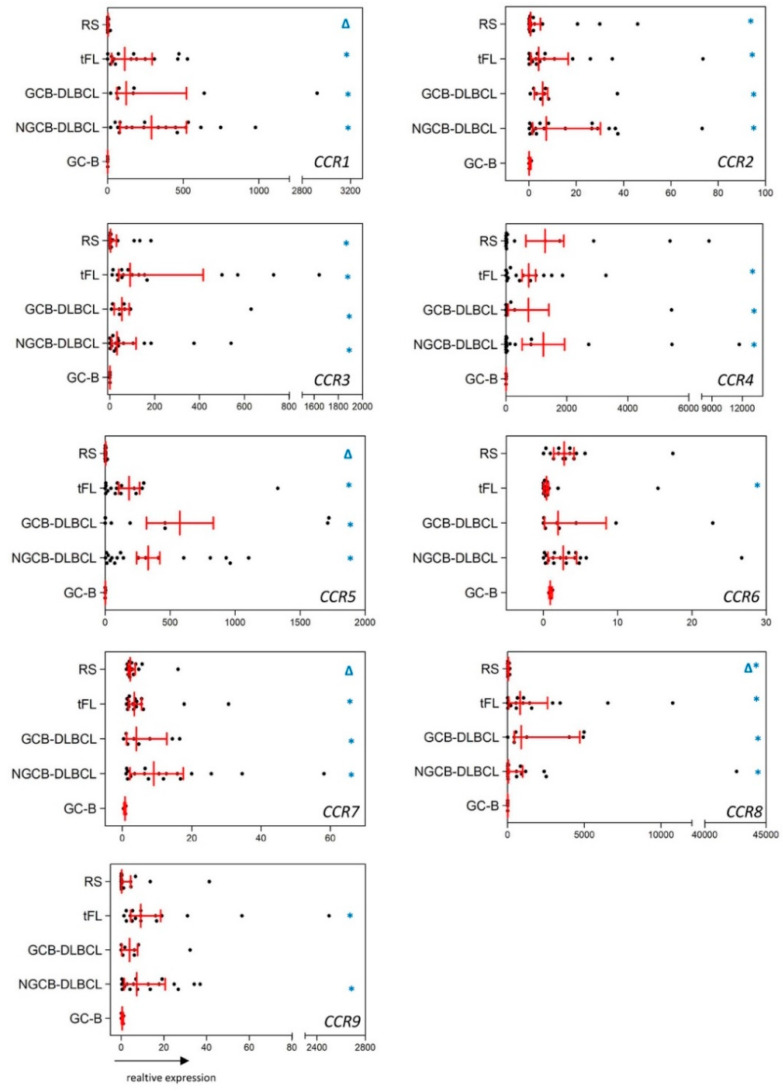
mRNA expression of CCR chemokine receptors in RS, tFL, GCB-DLBCL, NGCB-DLBCL, and GC-B as healthy controls. Scatter plots show the mRNA expression of *CCR1*, *CCR2*, *CCR3*, *CCR4*, *CCR5*, *CCR6*, *CCR7*, *CCR8*, and *CCR9* in RS, tFL, GCB-DLBCL, NGCB-DLBCL, and GC-B. GC-B denotes germinal center B-cells, which were isolated from non-neoplastic tonsils. Values of gene expression are calculated as relative expression. Red lines indicate median and interquartile range. * depicted in blue indicates significantly dysregulated chemokine receptors of a given lymphoma subgroup compared with GC-B (*p* < 0.05). ∆ depicted in blue indicates significantly dysregulated chemokine receptors of RS compared with the tFl, GCB-, and NGCB-DLBCL (*p* < 0.05).

**Figure 2 ijms-23-07874-f002:**
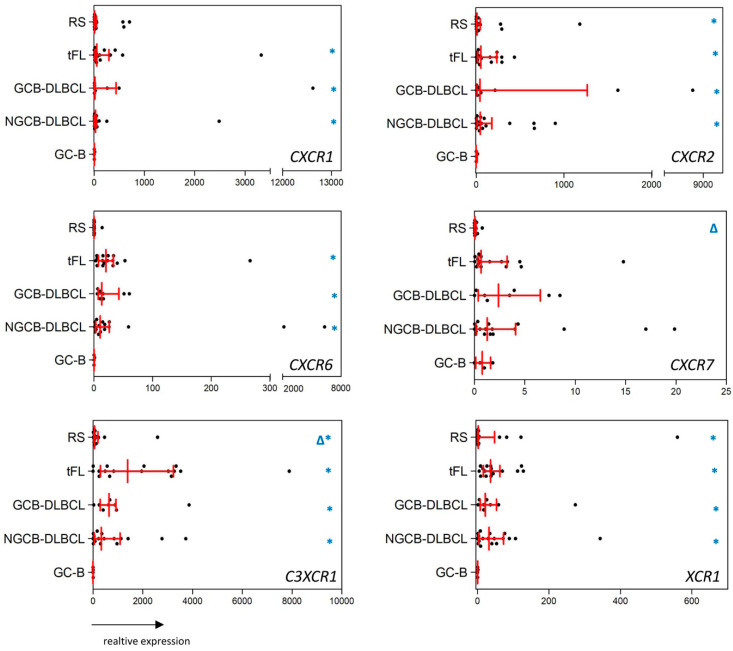
mRNA expression of CXCR, CX3CR1, and XCR1 chemokine receptors in RS, tFL, GCB-DLBCL, NGCB-DLBCL, and GC-B serving as healthy controls. Scatter plots show the mRNA expression of *CXCR1*, *CXCR2*, *CXCR6*, *CXCR7*, *CX3CR1*, and *XCR1* in RS, tFL, GCB-DLBCL, NGCB-DLBCL, and GC-B. GC-B denotes germinal center B-cells, which were isolated from non-neoplastic tonsils. Values of gene expression are calculated as relative expression. Red lines indicate median and interquartile range. * depicted in blue indicates significantly dysregulated chemokine receptors of a given lymphoma subgroup compared with GC-B (*p* < 0.05). ∆ depicted in blue indicates significantly dysregulated chemokine receptors of RS compared with the tFL, GCB- and NGCB-DLBCL (*p* < 0.05).

**Figure 3 ijms-23-07874-f003:**
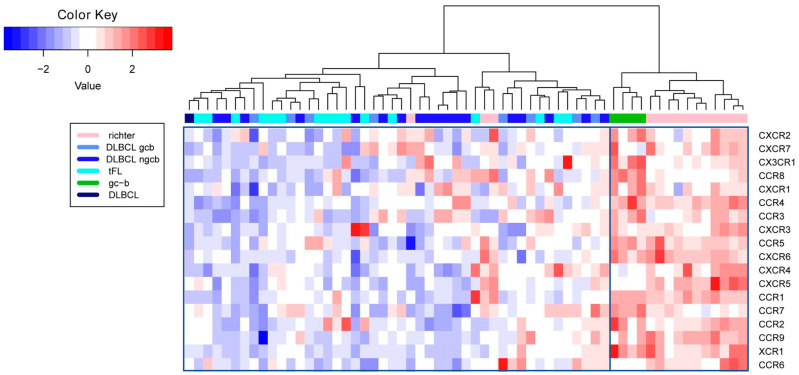
Heatmap based on ΔCT-values of the 18 analyzed chemokine receptors for 4 GC-B, 8 GCB--DLCBL, 18 NGCB-DLBCL, 16 tFL, and 14 RS samples, and hierarchical clustering of these analyzed samples. The color at the top designates whether the sample is a GC-B (green), RS (pink), GCB-DLBCL (light blue), NGCB-DLBCL (blue), tFL (bright blue), or unclassified DLBCL (dark blue) specimen. Scaled ΔCT-values are visualized in shades of blue to red (lower to higher values), with lower values representing a higher mRNA expression and higher values indicating lower mRNA expression.

**Figure 4 ijms-23-07874-f004:**
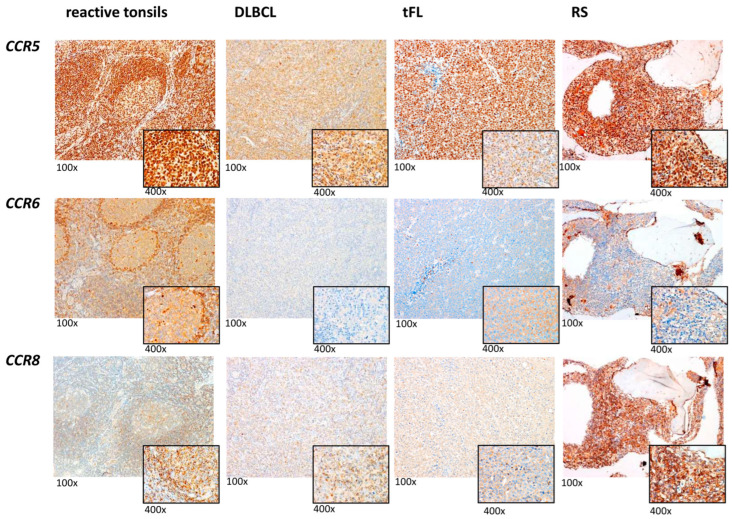
Representative immunohistochemical stains of CCR5, CCR6, and CCR8 expression in reactive tonsils, de novo DLBCL, tFL, and RS. All images were captured using an Olympus BX51 microscope and an Olympus E-330 camera (magnification 100× for the overview and magnification 400× for the small picture).

**Figure 5 ijms-23-07874-f005:**
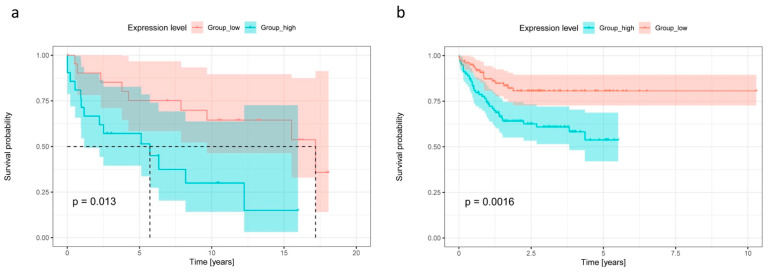
Overall survival in relation to chemokine receptor expression of the two aggressive lymphoma cohorts. Kaplan–Meier plot showing the overall survival of all de novo DLBCL and tFL patients, split into two groups based on the level of *CCR7* expression (high *CCR7* expression in blue, low expression in red, split by median). Survival curves of the de novo DLCBL and tFL cohorts are shown in (**a**). Survival curves of publicly available mRNA dataset Lenz et al. [21] (E-GEOD-10846 dataset available from ArrayExpress) is presented in (**b**).

**Figure 6 ijms-23-07874-f006:**
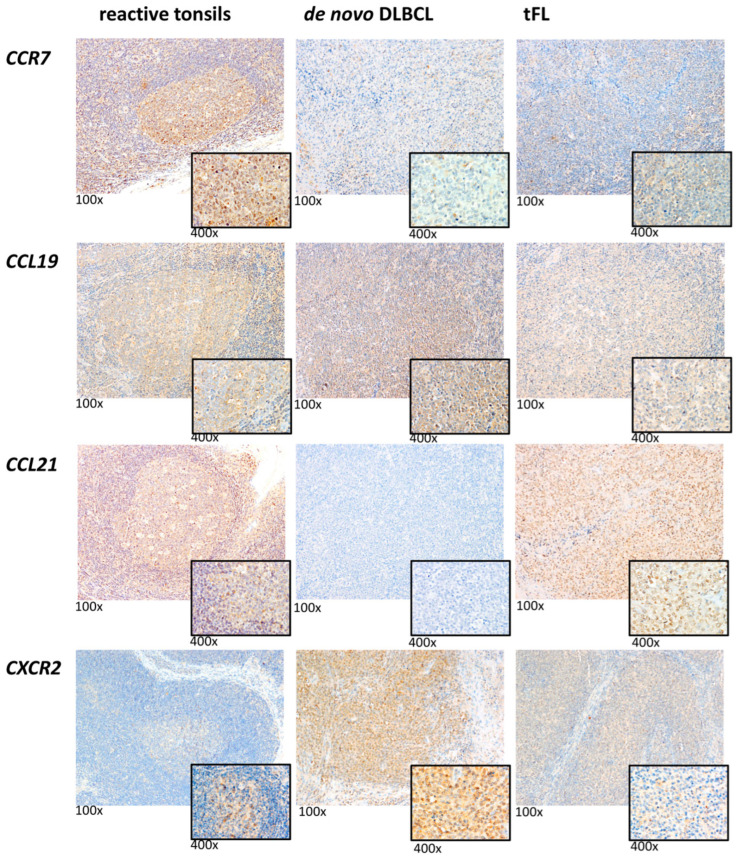
Representative immunohistochemical stains of CCR7, CCL19, CCL21, and CXCR2 expression in reactive tonsils, de novo DLBCL, and tFL. All images were captured using an Olympus BX51 microscope and an Olympus E-330 camera (magnification 100× for the overview and magnification 400× for the small picture).

**Table 1 ijms-23-07874-t001:** Analysis of intratumoral T-cells (CD3^+^), T helper (CD3^+^CD4^+^), cytotoxic T (CD3^+^CD8^+^) cells and macrophages (CD68^+^) determined by multicolor-immunofluorescence staining on selected cases.

Type of Lymphoma	CD3^+^ (Cells/mm²)	*p*-Value	CD3^+^CD4^+^ (Cells/mm²)	*p*-Value	CD3^+^CD8^+^ (Cells/mm²)	*p*-Value	CD68^+^ (%) *	*p*-Value
GCB-DLBCL	1491.7	>0.45	909.5	>0.8	499.7	>0.64	18.75	NGCB-DLBCL vs. tFL: 0.035
Range:	(760.6–2557.6)		(476.2–1828.4)		(275.3–1003.1)		(10–40)	
NGCB-DLBCL	1365.8		790.3		507.2		20	
Range:	(201.8–3493.3)		(60.1–2510.6)		(105.2–1184.5)		(5–40)	GCB-DLBCL vs. tFL: >0.38
tFL	1035.2		560.8		545.9		12.1	
Range:	(347.3–1605.3)		(121.8–1198.8)		(103.1–1138.5)		(5–20)	

* CD68^+^ (%) denotes the average percentage of CD68 positively stained cells per section.

**Table 2 ijms-23-07874-t002:** Mean of the percentage and immunoreactive score (IRS) of positive cells of CCR1, CCR4, CCR5, CCR6, and CCR8 in de novo DLBCL, GCB-DLBCL, NGCB-DLBCL, tFL, and RS.

	CCR1			CCR4			CCR5			CCR6			CCR8		
	% Cases (+)	%	IRS *-Mean	% Cases (+)	%	IRS *-Mean	% Cases (+)	%	IRS *-Mean	% Cases (+)	%	IRS *-Mean	% Cases (+)	%	IRS *-Mean
de novo DLBCL	30	7.1 (0–50)	0.7	30	19 (0–90)	2.4	100	90 (85–95)	16.2	80	57 (0–90)	6.6	90	67 (0–90)	11.7
GCB-DLBCL	0	0 (<0)	0	0	0 (<0)	0	100	90 (85–95)	18	50	25 (0–50)	2.5	100	70 (50–90)	7
NGCB-DLBCL	38	8.9 (0–50)	0.9	38	23.7 (0–5)	3	100	90 (85–95)	15.8	88	65 (0–90)	7.6	88	66.3 (0–90)	12.9
tFL	7	0.1 (0–1)	0.1	60	43.3 (0–90)	6.1	100	83.3 (30–90)	16.1	87	51.7 (0–90)	7	73	56.7 (0–90)	10.8
RS	0	0 (<0)	0	50	45 (0–90)	4.5	100	70 (30–90)	13	100	51.7 (0–90)	5.2	33	30 (0–90)	3

* IRS denotes the immunoreactive score determined using the percentage of positive cells and staining intensity as described in detail in the Materials and Methods section.

**Table 3 ijms-23-07874-t003:** Mean percentages and IRS of cells positive for CCR7, CCL19, CCL21, and CXCR2 in de novo DBLCL, GCB-DLBCL, NGCB-DLBCL, and tFL.

	CCR7			CCL19			CCL21			CXCR2		
	% Cases (+)	%	IRS-Mean	% Cases (+)	%	IRS-Mean	% Cases (+)	%	IRS-Mean	% Cases (+)	%	IRS-Mean
de novo DLBCL	100	74.5 (5–100)	12.63	100	75.2 (20–100)	9.36	32	24.7 (0–100)	3.69	88	72.5 (0–95)	5.44
GCB-DLBCL	100	75 (20–100)	13.5	100	85 (80–100)	7	17	13.3 (0–80)	2.67	92	63.78 (0–95)	2
NGCB-DLBCL	100	74.4 (5–100)	12.42	100	72.8 (20–100)	9.92	36	27.4 (0–100)	3.94	100	75.4 (40–95)	6.5
tFL	100	87.7 (20–100)	15.91	100	73.9 (10–100)	7.78	23	19.3 (0–100)	2.23	100	83.3 (60–95)	6.67

**Table 4 ijms-23-07874-t004:** Clinicopathologic characteristics of lymphoma patients included in this study.

	De Novo DLBCL		tFL		RS	
Clinicopathologic Parameters	Patients (*n*: 27)	Proportion (%)	Patients (*n*: 16)	Proportion (%)	Patients (*n*: 14)	Proportion (%)
* **Gender** *						
Male	15	55.6	5	31.25	10	71.4
Female	12	44.4	11	68.75	4	28.6
* **Age** *						
<=60	6	22.2	7	43.75	5	35.7
Male	5	18.5	4	25.0	4	28.6
Female	1	3.7	3	18.75	1	7.1
>60	21	77.8	9	56.25	9	64.3
Male	10	37.0	1	6.25	6	42.8
Female	11	40.7	8	50.0	3	21.4
* **Ann Arbor Stage** *						
1	7	25.9	2	12.5	n/a *	
2	8	29.6	2	12.5		
3	6	22.2	7	43.75		
4	4	14.8	4	25.0		
Not classifiable	2	7.4	1	6.25		
* **Immunophenotype of DLBCL (Hans algorithm)** *						
GCB	8	29.6	16	100	0	0
NGCB	18	66.7	0	0	5	35.7
Unclassifiable	1	3.7	0	0	9	64.3

Not classifiable: The respective parameters could not be determined in the indicated number of patients; the indicated proportion of a parameter is always referred to the total number of study patients for whom clinical data were available. Abbreviations: germinal center B-cell-like (GCB); non-germinal center B-cell-like (NGCB); transformed follicular lymphoma (tFL); Richter syndrome (RS); not applicable (n/a). * the Ann Arbor Stage was not evaluated in RS because of the pre-existing chronic lymphocytic leukemia.

## Data Availability

The public available data set used in this study can be found in the ArrayExpress database (https://www.ebi.ac.uk/arrayexpress/experiments/E-GEOD-10846/, accessed on 1 July 2022).

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
