# Peer review of "Distinct Chemokine Receptor Expression Profiles in De Novo DLBCL, Transformed Follicular Lymphoma, Richter’s Trans-Formed DLBCL and Germinal Center B-Cells"

_ijms, 2022, doi:10.3390/ijms23147874_

Round 1

Reviewer 1 Report

The authors want to investigate the expression profiles of chemokine receptors in different lymphoma and Germinal Center B-Cells. Some errors should be corrected.

1. On page 4, in the second paragraph, In the group of CXCR, CX3R1, and XCR1……. CX3R1 should be CX3CR1.

2. In the Figure 2 legend, mRNA expression of CXC, CX3CR1, and XCR1,….

The CXC should be CXCR

3. Figure 3, is the black frame should be shifted to the left side? In the result, the author descript that the heatmap can divide into two groups, but the black frame may be put in the wrong position.

4. In section 2.3., there are 16 de Novo DLBCL, why only use one sample to detect the mRNA levels of the chemokine receptor?

5. In figure 6, the GC-Bs express strong CCR7 proteins, but the figure 5, the worse overall survival is CCR7 strong expression, how to explain the result?

Author Response

Authors: In general, we thank the editor and all Reviewers for the detailed examination of our submitted paper and for their comments. All raised issues are addressed below, and all modifications are highlighted by using the track change mode in the resubmitted manuscript.

Reviewer 1

Comments and Suggestions for Authors

The authors want to investigate the expression profiles of chemokine receptors in different lymphoma and Germinal Center B-Cells. Some errors should be corrected.

Authors: Thank you for the detailed examination of our work, the positive remarks, and the suggestions for improvement.

  1. On page 4, in the second paragraph, In the group of CXCR, CX3R1, and XCR1……. CX3R1 should be CX3CR1.

Authors: We thank the Reviewer for this very important comment, and we have now corrected the error.

  1. In the Figure 2 legend, mRNA expression of CXC, CX3CR1, and XCR1,….

The CXC should be CXCR

Authors: We have corrected the figure legend accordingly to the Reviewer’s remark.

  1. Figure 3, is the black frame should be shifted to the left side? In the result, the author descrip that the heatmap can divide into two groups, but the black frame may be put in the wrong position.

Authors: Based on the Reviewer comments we have changed the Figure 3 of the manuscript.

  1. In section 2.3., there are 16 de Novo DLBCL, why only use one sample to detect the mRNA levels of the chemokine receptor?

Authors: In section 2.3, we described the immunohistochemistry and not the mRNA levels of 16 selected cases. For the mRNA levels we used 27 de novo DLBCL samples. For the immunohistochemistry we just depicted one representative image.

  1. In figure 6, the GC-Bs express strong CCR7 proteins, but the figure 5, the worse overall survival is CCR7 strong expression, how to explain the result?

Authors: We thank the Reviewer for raising this very important issue. Based on our findings altered CCR7 signaling (strong CCL19 expression whereas CCL21 is reduced in the lymphoma setting compared to the non-neoplastic controls), which we consider as possible explanation for the association of high CCR7 with poor outcome. Therefore, we implemented this comment in section 2.4.

Reviewer 2 Report

The authors analyzed mRNA levels for several cases of de novo DLBCL, tFL, Richter’s transformed DLBCL and compared with GC B cells. The research is solid and well performed. Some small remarks:

1.       What is the overlap between cases in all different analyses? Are cases used for T cell and macrophage infiltration selected from the cases in Table 4? And also for the IHC?

2.       In figure 1 the mean is used, while in some cases the mean seems very much affected by 1 outlier, with the distribution it makes more sense to use the median. The triangle placed if there is a significant difference with RS is placed at RS, that seems wrong!

3.       With the ranges in table 1 and the conclusion that the mRNA data must be of the B cells should there not be some kind of correlation done (if cases are the same at least)?

4.       The staining examples are not very clear, inseveral pictures the positive staining seems nuclear. Also the percentage of positive cells called in the tonsils (germinal center) seems not correct. The tFL case: are these FL part that is still present, seems not diffuse?

5.       Tables 2 and 3 are not clear, maybe a column with the percentage of positive cases can be added, the variation in % of positive cells is very high, is for the IRS  the mean or median used, not sure what you can do with a number with 2 decimals as the variation is high.

Author Response

Authors: In general, we thank the editor and all Reviewers for the detailed examination of our submitted paper and for their comments. All raised issues are addressed below, and all modifications are highlighted by using the track change mode in the resubmitted manuscript.

We thank the Reviewer for the detailed examination of our submitted paper and for the comments.

  1. What is the overlap between cases in all different analyses? Are cases used for T cell and macrophage infiltration seleced from the cases in Table 4? And also for the IHC?

Authors: Thank the Reviewer for this question/comment, in our analyses we used the identical samples for the analyses (RNA, IHC and TissueFAXS) as now mentioned in the different sections.

  1. In figure 1 the mean is used, while in some cases the mean seems very much affected by 1 outlier, with the distribution it makes more sense to use the median. The triangle placed if there is a significant difference with RS is placed at RS, that seems wrong!

Authors: Thank the Reviewer for this comment, we took this feedback in concern and changed Figure 1 and Figure 2. Now we are using box plots to better present the data (median and interquartile range). Furthermore, the error was now corrected.

  1. With the ranges in table 1 and the conclusion that the mRNA data must be of the B cells should there not be some kind of correlation done (if cases are the same at least)?

Authors: A correlation has been performed as we already stated in section 2.2 (“No correlation between the number of intratumoral T-cells or their subpopulations, macrophages, and the chemokine receptor profile was observed, suggesting that the chemokine receptor profile might be determined by the presence of lymphoma B-cells and not by the infiltrating non-malignant immune cells.”) and we found no correlation for the CCR expression profile and the immune cell infiltrates.

  1. The staining examples are not very clear, in several pictures the positive staining seems nuclear. Also, the percentage of positive cells called in the tonsils (germinal center) seems not correct. The tFL case: are these FL part that is still present, seems not diffuse?

Authors: Because this raised issue, we re-evaluated all IHC stains once more together with our pathologist: In some cases, strong expression could be misinterpreted as nuclear expression. Furthermore, we changed the images of tFLs, because as stated by the reviewer, the used sample really seemed that the FL part is still present and now we used a different part of the tFLs showing the high-grade component. Concerning the percentage of germinal center cells (centroblasts and centrocytes), the given percentage is correct based on the evaluation of our pathologist.

  1. Tables 2 and 3 are not clear, maybe a column with the percentage of positive cases can be added, the variation in % of positive cells is very high, is for the IRS the mean or median used, not sure what you can do with a number with 2 decimals as the variation is high.

Authors: Thanks for the very important issue. We now implemented the percentage of positive cells in the two tables in a separate column. Furthermore, we added mean to the IRS in the header and we depicted with just one decimal.